# Musical instrument classifier for early childhood percussion instruments

**Brandon Rufino**[1,2], **Ajmal Khan**[1☉], **Tilak Dutta**[2,3☉], **Elaine Biddiss**[1,2,4]*

**1** Bloorview Research Institute, Holland Bloorview Kids Rehabilitation Hospital, Toronto, Ontario, Canada, **2** Institute of Biomedical Engineering, University of Toronto, Toronto, Ontario, Canada, **3** KITE, Toronto Rehabilitation Institute, University Health Network, Toronto, Ontario, Canada, **4** Rehabilitation Sciences Institute, University of Toronto, Toronto, Ontario, Canada

☉ These authors contributed equally to this work.
* ebiddiss@hollandbloorview.ca

**Data Availability Statement:** The data that support the findings of this study may be made available on request from the corresponding author, E.B, in compliance with institutional and ethical standards of operation. Data cannot be shared publicly because research participants did not provide

## Abstract

While the musical instrument classification task is well-studied, there remains a gap in identifying non-pitched percussion instruments which have greater overlaps in frequency bands and variation in sound quality and play style than pitched instruments. In this paper, we present a musical instrument classifier for detecting tambourines, maracas and castanets, instruments that are often used in early childhood music education. We generated a dataset with diverse instruments (e.g., brand, materials, construction) played in different locations with varying background noise and play styles. We conducted sensitivity analyses to optimize feature selection, windowing time, and model selection. We deployed and evaluated our best model in a mixed reality music application with 12 families in a home setting. Our dataset was comprised of over 369,000 samples recorded in-lab and 35,361 samples recorded with families in a home setting. We observed the Light Gradient Boosting Machine (LGBM) model to perform best using an approximate 93 ms window with only 12 mel-frequency cepstral coefficients (MFCCs) and signal entropy. Our best LGBM model was observed to perform with over 84% accuracy across all three instrument families in-lab and over 73% accuracy when deployed to the home. To our knowledge, the dataset compiled of 369,000 samples of non-pitched instruments is first of its kind. This work also suggests that a low feature space is sufficient for the recognition of non-pitched instruments. Lastly, real-world deployment and testing of the algorithms created with participants of diverse physical and cognitive abilities was also an important contribution towards more inclusive design practices. This paper lays the technological groundwork for a mixed reality music application that can detect children's use of non-pitched, percussion instruments to support early childhood music education and play.

## Introduction

Musical play and learning, particularly prior to 7 years of age, can enhance outcomes in self-efficacy, perception, fine motor coordination and spatial reasoning [1]. Yet, there are

consent for public sharing of their data. To ensure the long-term stability and accessibility of our research data, we will designate a non-author institutional contact, the research ethics committee chair. This approach ensures that the data remains accessible over time, providing a reliable point of contact for interested researchers. Such an arrangement is particularly beneficial in cases where an author may change their email address, shift to a different institution, or become unavailable to respond to data access requests. Please see the contact information for the non-author institutional contact below: Deryk Beal Research Ethics Board Chair Holland Bloorview Kids Rehabilitation Hospital 150 Kilgour Road, Toronto, ON M4G 1R8 Tel: (416) 425-6220, ext.3582 E-mail: dbeal@hollandbloorview.ca.

**Funding:** The authors would like to acknowledge research funding provided through the Collaborative Health Research Projects program (FRN 163978) by the Natural Sciences and Engineering Research Councils of Canada, the Canadian Institutes of Health Research, and the Social Sciences and Humanities Research Council of Canada (https://www.nserc-crsng.gc.ca/professors-professeurs/grants-subs/chrpprcs_eng.asp), as well as the Queen Elizabeth II-Graduate Scholarship in Science and Technology program funded through the Ontario provincial governments (https://osap.gov.on.ca/OSAPPortal/en/A-ZListofAid/PRDR019236.html). The authors would also like to thank Rhythm Band Instruments (TX, USA) for their generous donation of instruments in support of this research. We are also extremely grateful to the Ontario Brain Institute for funding and support through the integrated discovery program, CP-Net (http://cpnet.canchild.ca/). We would also like to thank Scotiabank for its generous support of the gaming hub at the Bloorview Research Institute. The funders had no role in study design, data collection and analysis, decision to publish, or preparation of the manuscript.

**Competing interests:** I have read the journal's policy and the authors of this manuscript have the following competing interests: Holland Bloorview is supporting the creation of a company called Pearl Interactives to commercialize products like Bootle Band so that it can be made widely available to those who can benefit from it. Elaine Biddiss and Ajmal Khan are shareholders in Pearl Interactives and may financially benefit from this interest if Pearl Interactives is successful in marketing products related to this research including Bootle Band. The terms of this arrangement have been reviewed and approved by Holland Bloorview Kids

significant opportunity gaps when it comes to early childhood music education [2–4]. Children with motor disabilities and those from low-income households are much less likely to learn to play a musical instrument and participate in early childhood music learning programs [5–7]. Barriers to early childhood music education include finding suitable programs [2], costs [3], travel constraints and time [4]. These barriers are not easily overcome by traditional teacher-led programs. Music applications "apps" may offer opportunities to close opportunity gaps in early childhood music education for children with and without disabilities. However, a review of some of the most popular mainstream music apps for young children concluded that they generally do not promote diverse, frequent, and active music engagement [8]. Active engagement, wherein real-life instruments are manipulated introducing sensory and physical task parameters, is vital to music and motor learning [9].

"Mixed reality" apps have recently emerged that detect and respond to audio signals from real-life, pitched instruments (e.g., piano, violin, guitar) [10]. However, in early childhood, non-pitched percussion instruments (e.g., shakers/maracas, tambourines and castanets) are more prominent and appropriate for developing motor abilities, especially for children with disabilities who may have limitations in fine motor control [11, 12]. Musical instrument and audio classifiers have been built with a wide range of machine learning models including k-nearest neighbors (KNN) [13], Multi-layered Perceptron (MLP), and boosting algorithms [14]. Harish et al. reported accuracies of 79% for their SVM model which outperformed the other state-of-the-art models with in classifying six pitched instruments, including voice, using spectral features [14]. Mittal et al. demonstrated best performance with a Naive Bayes Classifier with an accuracy of 97% for distinguishing between 4 drum instruments using a dataset composed of both live recordings as well as a drum simulator [15]. While the musical instrument classification task for pitched instruments and drums are widely studied [16], we were not able to find previous work classifying diverse non-pitched instruments like maracas, tambourines, and castanets. Classification of non-pitched instruments may pose additional challenges due to greater overlaps in frequency bands and variation in sound quality and play style than pitched instruments [17].

Pairing signal processing techniques with machine learning offers opportunities to address current limitations in the detection of non-pitched percussion instruments. Improving signal robustness, increasing recognition of families of instruments (e.g., any shaker regardless of size and material), and prioritizing computationally low-cost approaches are important pragmatic considerations for the development of classifiers that will enable children to play with the instruments they have on hand [18] and with technology they have in their home (i.e., a tablet, computer, or phone).

Another key consideration for designing an instrument classifier for use by children in a home-based music application is the complex sound environment. For instance, in our target application, the audio stream may include speech, background noise, game/app music, as well as sounds from the musical instruments of interest. There are several approaches to the classification task for polyphonic audio (i.e., audio with multiple sources of sound) [19–22]. However, the most common and straightforward approach is to extract features directly from the polyphonic signal to classify the musical instrument [23, 24]. This last approach minimizes computation and latency, which are important practical considerations for a classifier intended for a music application.

In this work, we aimed to develop an audio detection interface for non-pitched percussion instruments, specifically maracas, tambourines, and castanets, for use in early childhood music applications. To this end, this manuscript first describes the creation of a large database of non-pitched instrument audio samples. Second, we describe feature extraction and the development of machine learning models used in this classification task. Next, we present the

Rehabilitation Hospital and the University of Toronto in accordance with its policy on objectivity in research. We will continue to actively monitor, mitigate and manage any conflicts of interest. Our goal is to remain transparent and committed to the best interests of study participants, patients and families. This does not alter our adherence to PLOS ONE policies on sharing data and materials.

performance of our classifier with (i) a test set recorded in-lab and, (ii) real-world data recorded "in the wild" in family homes. Lastly, we discuss key findings, particularly with respect to the algorithms intended application as an audio detection interface to support interactive early childhood music applications.

## Methodology

### Dataset

**1. Non-pitched percussion dataset.** Given the unavailability of databases with well-labelled samples of diverse, non-pitched percussion instruments, a novel dataset was created for this purpose consisting of four classes: (1) tambourines, (2) maracas/shakers, (3) castanets, and the (4) noise class (e.g., white noise, people speaking, environment sounds etc.). This dataset (described in Table 1) was accumulated over June 2019 to May 2021 in various locations (e.g., family homes, coffee shops, outdoors and basements) within Ontario, Canada. It will be referred to as the "in-lab" dataset to convey that the recordings were made in contrived rather than naturalistic conditions. Audio samples were recorded using a single microphone channel (i.e., mono) at 44.1 kHz. Most samples recorded are polyphonic (are mixed with people speaking, coffee shop sounds like the sound of cutler, different frequencies of white noise and background music) to reflect the anticipated sound environment of the target application.

When recording samples for each instrument class, a variety of instruments was used to avoid bias in the brand and material of the instrument. For example, egg shakers, wooden maracas and plastic maracas were recorded for the shaker class. For the tambourine class, we used tambourines varying in diameter from 8 cm to 30 cm and of different materials such as cowhide, plastic and wood. For the castanet, hand-held and finger castanets were sampled—some of which were made from plastic and others from wood. For all three instrument classes, homemade instruments were also included. Our goal in doing so was to alleviate any financial barriers a family may face when taking part in musical play and learning at home given that not all families may have access to musical instruments. "Do It Yourself" (DIY) instructions were created for each instrument class using common household items. A homemade version of a shaker was constructed from a plastic water bottle partially filled with rice. For a homemade tambourine, we trialed a ring of three keys or a paper plate with four metal jingle bells attached around the circumference. Lastly, a homemade castanet was simulated using two wooden spoons, or a folded paper plate with approximately four coins glued onto each side. See the supporting information file (S1 Appendix) for an image of the homemade instrument options.

**2. Dataset splits.** For training and evaluating the classifier to the "in-lab" dataset, we split our corpus into approximately 80/10/10% train, validation and test set. The validation set included instruments and ambient sound environments that were not part of the training set. Of note, we labelled and grouped by instruments before data splitting to ensure that there was

**Table 1. Data splits for "in-lab" dataset: training, validation and testing.** Samples calculated with ≈93 ms window and 50% overlap.

| Class | Training Samples | Validation Samples | Test Samples ("in-lab") variants |
|---|---|---|---|
| Tambourines | 88,339 | 12,217 | 11,491 |
| Shakers | 93,735 | 15,102 | 10,968 |
| Castanets | 34,363 | 9,773 | 4,102 |
| Noise | 73,207 | 8,506 | 7,273 |
| Total | 289,644 | 45,598 | 33,834 |

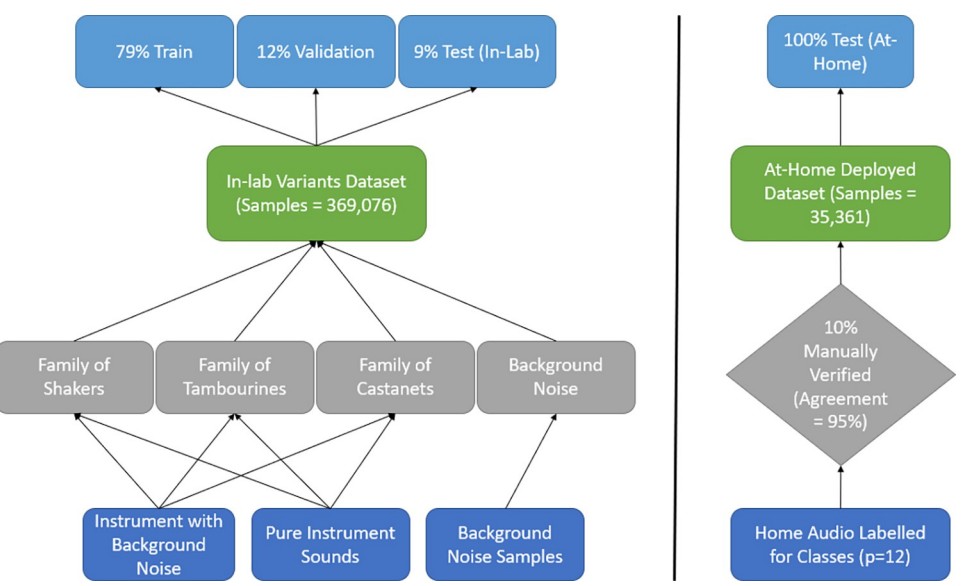

**Fig 1. Data collection summary including in-lab and at-home deployed samples.**

no overlap in samples between the training and testing sets. The test set fully comprised of either (1) instruments that were not included in training and validation, or (2) instruments that were included in training/validation but recorded in a totally new environment. This prevented the classifier from being biased to certain brands and materials of instruments. Table 1 summarizes the data splits and Fig 1 summarizes the complete data collection process, including the at-home deployment test set (to be discussed in Classifier Evaluation).

## Features and window length

**1. Windowing length.** The general approach for musical instrument classification is to discretize the signal based on a window time length and extract relevant features [16]. There are no established standards for windowing time for the musical instrument classification task. Previous successful work has used window lengths between 24–500 ms and 50% overlap between the windows [25, 26]. For this work, a maximum window length of ≈93 ms was specified because tolerable latency in gameplay is reported to be under 100 ms [27]. A longer window length would also increase the likelihood of multiple instruments being played within the window or the signal not being stationary (a key assumption for our features). As the Fast Fourier Transform (FFT) [28] was used, a sample length to a power of two was maintained to minimize the computation time. Using a common audio recording sample rate of 44.1 kHz, we therefore experimented with the following window lengths: ≈23 ms windows (1024 samples), ≈46 ms windows (2048 samples) and ≈93 ms windows (4096 samples).

**2. Feature extraction and selection.** Dimensionality for the musical instrument classification task ranges widely [29] with previous work using anywhere from 27 [30] to 162 time and spectral features [29]. A full table of the features that extracted in this work can be found in the supporting information file (S2 Appendix) from which neighborhood component analysis (NCA) was used to select an optimal feature set. NCA is a non-parametric method with the goal of maximizing accuracy of a classification algorithm [31]. We used the fscna tool provided by Matlab [32] which performs NCA with a regularization term. The first 12 mel-frequency cepstral coefficients (MFCCs) and signal entropy were found to contribute most to the model across all windowing times. Below, we detail methods associated with obtaining these features.

*MFCCs*. MFCCs are used to represent shapes of sounds and are calculated using standard procedures [33]. The bandwidth of interest and the number of mel-filterbanks used must be specified in line with sampling rates [25, 34, 35]. A maximum frequency of 8 kHz was selected as a conservative approach in case our classifier is ever deployed to a hardware device that is limited to a sampling rate of 16 kHz (albeit, if the classifier was deployed to a 16 kHz device, it should be retrained using a down sampled version of our dataset). For the interested reader, spectrograms of each instrument are presented in the supporting information file (S3 Appendix). Music classification uses anywhere from 20–40 mel-filterbanks to summarize the power spectrum [34]. As is typical, we discarded the first MFCC because it is related to only the signal energy and kept the subsequent 2–20 MFCCs [25].

*Signal entropy (also referred to as Shannon's Entropy [36]).* Musical instruments like castanets with a sharp signal onset and decay will contain more information (entropy) than a flat white noise frequency in the background. Signal entropy was calculated using equations provided in previous work [36]. To avoid underestimation, a bias correction term was applied, following the work by Moddemeijier [37].

## Classifier development

We developed models using KNN, SVM, MLP and the boosting algorithms, AdaBoost [38], XGBoost [39] and Light Gradient Boosting Machine (LGBM) methods [40]. We hyper-parameter tuned the LGBM models using Optuna [41]. Optuna offers a flexible API along with a sampling algorithm to efficiently optimize the search space. We report the optimal parameters and other baseline model performances for the ≈ 93ms window in the supporting information file (S4 Appendix). Fig 2 presents an overview of the model development and evaluation process.

## Classifier evaluation

**In-lab test phase.** Accuracy, recall, precision and F1 scores were calculated to quantify classifier performance and compared between models using window lengths of ≈23 ms, ≈46 ms, and ≈93 ms.

**Real-world deployed phase.** The following describes the methodology associated with the real-world deployment phase to evaluate the performance of the classifier in the naturalistic

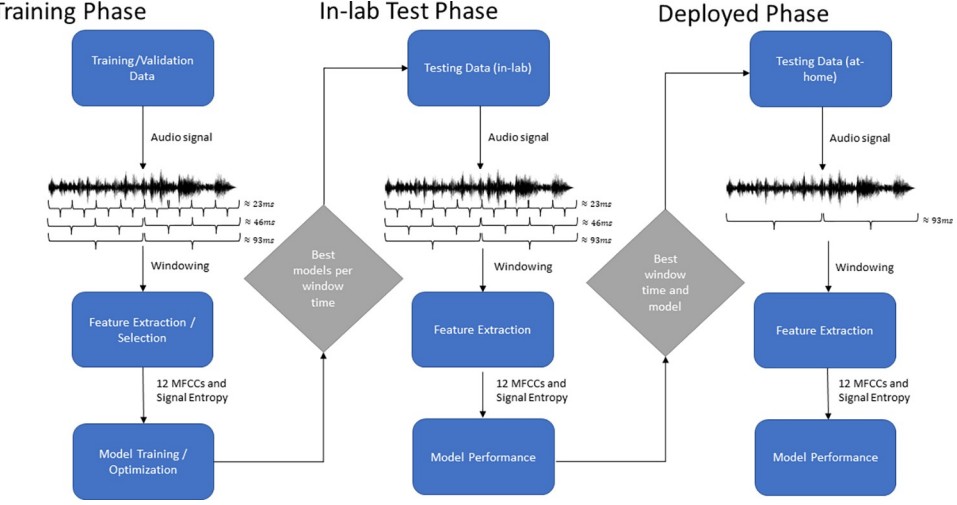

**Fig 2. Development and evaluation of non-pitched percussion musical instrument classifier.**

home setting. This was considered an important step to identify any biases in our model due to the variation in environment, play styles and instruments and to ascertain its practical viability. In-home test data were collected in the context of a usability study wherein the classifier was embedded within an existing game application, Bootle Band.

*Materials.* Bootle Band is an early childhood music application developed at the Possibility Engineering And Research Lab (PEARL) at Holland Bloorview Kids Rehabilitation Hospital and available on the Apple App Store. In Bootle Band, children play their musical instrument to interact with cartoon characters and advance the narrative. Families were provided an iPad with Bootle Band installed as well as a set of musical instruments that included a maraca, an egg shaker, a castanet, and a tambourine (Rhythm Band Instruments, Texas, US). Video instructions for the DIY instruments were also provided within the application. The research version of Bootle Band has the flexibility to be used as a data collection platform as audio and video data can be recorded during play with the family's permission.

*Participants.* Twelve families were recruited through Holland Bloorview Kids Rehabilitation Hospital and included 9 boys and 3 girls, with and without motor disabilities aged from 4 to 9 years (see Table 3 for a list of diagnoses represented in this study). The recruitment period spanned from February 22nd, 2021 to May 1st, 2021. Ethical approval for this study was obtained from the Holland Bloorview Research Ethics Board and the University of Toronto Health Sciences Research Ethics Board. Participants and/or their guardians provided informed and written consent using approved e-consent procedures facilitated by REDCap. Children who were unable to consent provided assent.

*Protocol.* Families played Bootle Band with the audio detection interface for 2 weeks at home. The game prompted the family to adjust the volume of the iPad to a comfortable level before beginning to play but did not place any restrictions to noise that may be caused from external factors. During play, audio data were recorded with the family's permission. Roughly 27 minutes of audio data were collected. See Fig 3 for a dataset breakdown per participant. We also conducted an interview with the families at the end of the study and asked them if they observed any technical difficulties with the detection of the instruments.

*Analysis.* A researcher reviewed the recordings from the play sessions and manually identified the 4 classes of sound (i.e., shaker, tambourine, castanet or noise class). Ten percent of the labels were randomly reviewed by another researcher for accuracy and showed 95% in agreement using Cohen's kappa method [42]. Once labelled, we evaluated the performance of our classifier in the real-world home setting.

## Results

### In-lab performance

The LGBM classifier performed best on our validation set across all window lengths and for all performance metrics (Precision: 0.845; Recall: 0.835, F1: 0.839, Accuracy: 0.844, 93ms window). The AdaBoost model had similar performance (Accuracy: 0.837, 93ms window) with the logistic regression model (Accuracy: 0.676, 93ms window) and the SVM (Accuracy: 0.756, 93ms window) being the lowest performing. Henceforth, results reported are with reference to the LGBM model only. The interested reader is referred to the Supporting Information (S5 Appendix) for a detailed performance comparison of the different machine learning models.

Table 2 presents the performance of the LGBM model on the held-out "in-lab" test set. Using a ≈23 ms (1024 sample) window length we see an overall accuracy of about 72.6%. This accuracy increases when we expand our window to ≈46 ms (2048 samples) and ≈93 ms (4096 samples). Respectively, we see an increase in accuracy of 1.4% (to a total of 74.0%) and an

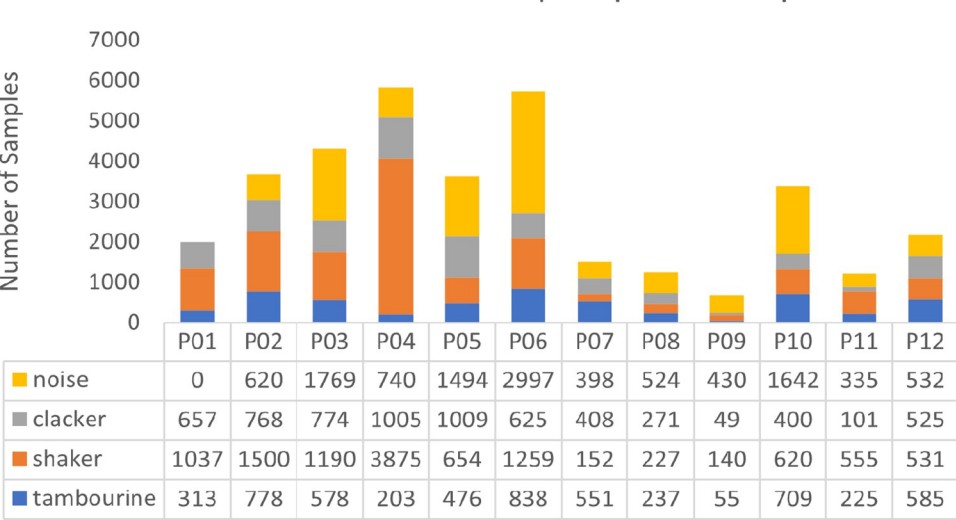

**Fig 3. At-home deployed dataset from a usability study with 12 families (≈ 93 ms window and 50% overlap).**

increase of 11.8% (to a total of 84.4%) from our smallest window size of ≈23 ms. Another substantial increase in performance can be observed in the precision of our classifier. For example, the lowest precision score was in the Tambourine class with a ≈23 ms window at 67.5%. This precision improved to 86.7% when the window size was increased to 93 ms. With this information, we deployed the classifier trained from the ≈93 ms window into Bootle Band for real-world playtime evaluation. The confusion matrix analysis for the LGBM model with a window size of 93 ms is provided in Supporting Files (S6 Appendix) for the interested reader.

## Real-world performance

Twelve participants aged 4 to 9 years with a range of diagnoses affecting motor abilities participated in the home trial with detailed demographics reported in Table 3.

Fig 4 summarizes the performance of our classifier against the real-world data collected as children played the game at home.

There was a significant decrease in accuracy across all instrument classes from the performance seen with the "in-lab" test set. When evaluated with the real-world data collected in family homes, the average accuracy decreased approximately 11% to 73.3% [SD = 6.8%]. We also observed a significant decrease in performance across all other macro averages (recall, precision and F1 score). The largest drop in performance was seen in the precision metric which decreased about 14% (from 84.5% [SD = 0.6%] to 70.5% [SD = 5.6%]). The smallest reduction in performance was seen in the recall metric with a decrease of 8.6% (from 83.5% [SD = 0.6%] to 74.9% [SD = 5.7%]).

Looking into individual instrument classes, we see a significant reduction in performance in recall for the noise class (69.9% median recall for at-home variants vs. 90.5% median recall for in-lab variants). The tambourine class showed the worst precision out of all 4 classes with a precision median of 63.5%. The tambourine also showed the lowest F1 score with a median of roughly 62.8%. All three instruments, apart from the noise class, demonstrated above 75% recall and the decrease in performance was not statistically significant from the in-lab variants.

In terms of variation from one household to the next, we observed that four participants had below 70% accuracy (P01 = 67.0%, P03 = 65.5%, P06 = 68.2% and P11 = 68.5%). We also

**Table 2. Non-pitched percussion instrument classifier to our "in-lab" test set against different window lengths.**

| Window Length (ms) | Instrument Class | Accuracy | Precision | Recall | F1 Score |
|---|---|---|---|---|---|
| ≈23 | Tambourine | | 0.6750 | 0.7285 | 0.7007 |
| | Shaker | | 0.7185 | 0.7801 | 0.7480 |
| | Castanet | | 0.8909 | 0.5181 | 0.6552 |
| | Noise | | 0.7158 | 0.8812 | 0.7900 |
| | Macro Average | 0.7255 | 0.7500 | 0.7270 | 0.7235 |
| ≈46 | Tambourine | | 0.6822 | 0.7480 | 0.7136 |
| | Shaker | | 0.7433 | 0.7963 | 0.7689 |
| | Castanet | | 0.8886 | 0.5303 | 0.6642 |
| | Noise | | 0.7304 | 0.8834 | 0.7997 |
| | Macro Average | 0.7395 | 0.7611 | 0.7395 | 0.7366 |
| ≈93 | Tambourine | | 0.8673 | 0.8063 | 0.8357 |
| | Shaker | | 0.8175 | 0.8781 | 0.8467 |
| | Castanet | | 0.8388 | 0.7512 | 0.7925 |
| | Noise | | 0.8553 | 0.9051 | 0.8795 |
| | Macro Average | 0.8441 | 0.8447 | 0.8352 | 0.8386 |

observed one outlier with an accuracy of 91.5% accuracy (P04). All other participants (n = 7) showed accuracies within the 70–80% range. To coincide with the quantitative analysis, when families were asked "Did you experience any technical difficulties with the detection of musical

**Table 3. Participants demographics (n = 12).**

| Demographic | Total participants |
|---|---|
| *Diagnosis* | |
| Cerebral Palsy, GMFCS < = Level 3 | 3 |
| Spinal Cord Injury | 1 |
| Hard of hearing | 1 |
| Hydrocephalus | 1 |
| MYBPC2 | 3 |
| Hemiparesis | 2 |
| Polymicrogyria | 1 |
| ADHD | 1 |
| No reported diagnosis | 4 |
| *Average annual family income* | |
| <$24,999 | 2 |
| $25,000 to $49,999 | 3 |
| $50,000 to $74,999 | 0 |
| $75,000 to $99,999 | 1 |
| $100,000 to $149,000 | 2 |
| >$150,000 | 2 |
| Prefer not to disclose | 2 |
| *Ethnicity* | |
| White | 3 |
| Hispanic | 1 |
| Black | 5 |
| Asian | 1 |
| Mixed heritage | 2 |

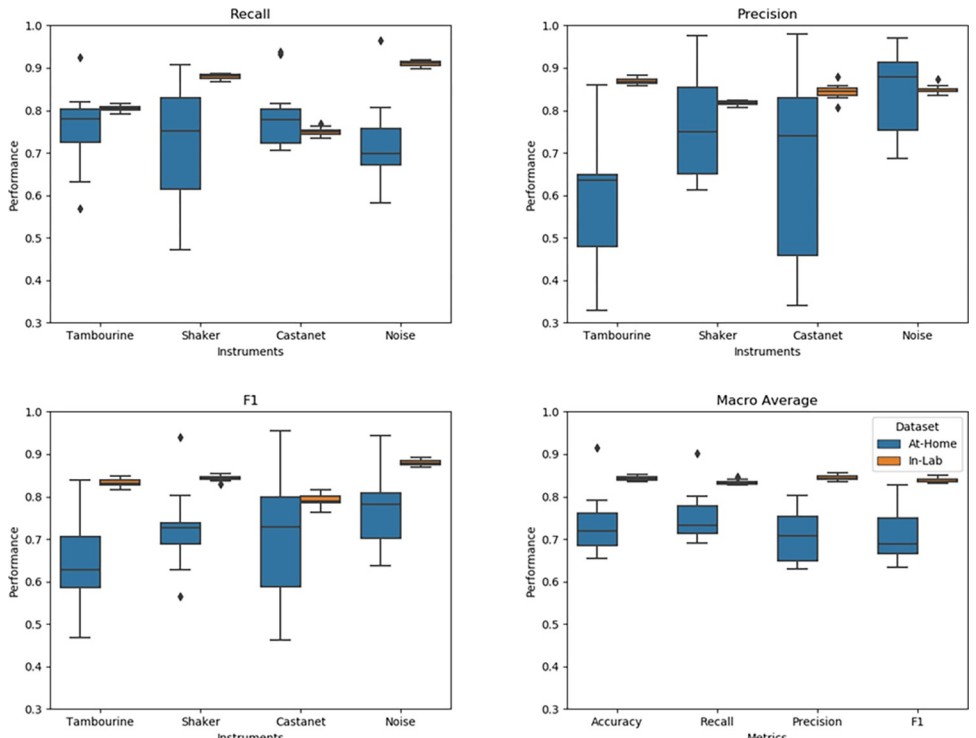

**Fig 4. Non-pitched percussion musical instrument classifier performance to real-world playtime setting (participants = 12).**

instruments?", eleven out of twelve families reported no latency or detection issues with Bootle Band. One out of the twelve caregivers noticed a very minor delay when an instrument was played vs. the reaction time of the game. However, the caregiver noted that their child never seemed to notice and it did not impede the play experience.

## Discussion

The goal of this research was to design an audio detection interface for non-pitched percussion instruments common in early childhood to support musical play and learning opportunities through low-cost, home-based applications. To this end, we created a large dataset of diverse audio samples which was used to design and evaluate a classifier which achieved 84.4% accuracy across 4 classes (shakers, castanet, tambourine, noise).

This shows that a machine learning classifier can find predictive patterns that are unique to families of percussion instruments. As anticipated, a reduction in performance of about 11% was observed when the classifier was deployed in a naturalistic home setting, however the performance was sufficient to sustain user satisfaction across all families. During this research, we prioritized the development of a technological solution that would translate well to real-world implementation and this focus shaped many of our design decisions. We also prioritized equity, diversion and inclusion in our research by including diverse children with and without disabilities in the home study in an effort to ensure that this research is translatable to families most in need of low-cost, early childhood music applications. Key findings are as follows:

**1) Multiple strategies are needed to address the high level of variability associated with non-pitched percussion instruments.** Within the same class of instrument, there exists a lot of variation from one instrument to another depending on the material of fabrication, size etc.

For example, a 12" cowhide tambourine purchased from a music store is sonically quite different from a 4" plastic tambourine purchased at a local toy store. To address this variability, we built a large dataset containing over 369,000 samples of diverse instruments recorded in varied contexts with which to train our classifier. While we attempted to capture a wide variety of instruments in each class, our data set was not exhaustive and will need continued expansion and iteration to capture the range of variability associated with non-pitched percussion instruments. To address this limitation, we conducted an error analysis of our classifier to understand which instruments were more likely to be inaccurately classified. First, we observed that there were frequent misclassifications between tambourines and shakers likely owing to an overlap of frequencies between these two instruments. When this behavior was explored in greater depth, we identified that plastic tambourines were often misclassified as shakers (compared to tambourines that used jingle bells or were made from wood/cowhide) and wooden shakers were more often misclassified than plastic shakers. These observations were important for guiding successful real-world deployment and recommendations for preferred instruments for use in the game. Another step that was taken to mitigate this limitation was the creation of the "DIY" instrument guides which resulted in low-cost instruments that could be consistently detected by our classifier with an average accuracy of 92% for the in-lab test set. Understanding the limitations of our classifier enabled us to generate and test these mitigating strategies, as well as guide future developments.

**2) Good classifier performance with low latency and computational demands typical of common mobile devices could be achieved using a small feature set.** We found that only a handful of features (MFCCs and signal entropy) were needed to classify percussion instruments in a short window of time (≈93 ms). By minimizing our features, we were able to reduce bias in our model and minimize computation time. This is crucial for our target application which required latency of less than 100 ms to ensure a good user experience [27]. While larger window times were associated with an increase in classifier performance, it is unlikely that this would have translated to a better user experience when embedded into a gaming application. Using the high speed SHAP algorithm [43, 44], we identified the features which contributed the most to our LGBM classifier. SHAP combines local explanations and optimal credit allocation using the classic Shapley values from game theory and their related extensions. We found that the first 4 MFCCs and signal entropy were the main drivers for our predictions. This finding makes sense since the first few MFCCs summarize most of the variance within the spectral envelope. Looking at signal entropy, we observed that it plays an important role for the castanet and noise predictions. We believe this is because the castanet family has a very sharp onset and decay compared to other instruments like a tambourine or a shaker, thereby producing a distinct entropy value. The opposite is true for noise, where most samples carry little information which is distinct from the other three families. Our SHAP results can be found in the supporting information file (S7 Appendix).

**3) "In the wild" evaluation is essential to characterizing system performance.** Our classifier achieved 84.4% accuracy across all three instrument families when evaluated with our in-lab test set with ≈93 ms windows. In deployment, it achieved 73.3% accuracy across all three instrument families. In the home setting, misclassifications could be attributed to many different events: children yelling, the microphone being covered when a child holds the iPad, and children playing two different instruments at the same time with their sibling/parent. Upon exploratory investigation we noticed that when the child was playing Bootle Band in a quiet environment without many interruptions like the ones listed above, the classifier performed with near equivalence to in-lab testing. Understanding the source of errors enables opportunities to provide families with better instructions for an optimal user experience as well as to inform technological developments that could help to mitigate misclassification (e.g.,

programmatic volume controls associated with the game music, warnings if the microphone is covered). As an example of the latter, one feature that we implemented to support deployment is that we allowed users, through a friendly front-end user interface, to toggle the probability threshold of predicting an instrument. For example, if a family was playing in a loud environment, then to reduce the number of false positives we allowed them to increase the probability threshold of our classifier to a max of 0.9. If the user were playing in a quiet environment, they could lower the probability threshold to 0.4. Doing this allowed the family to adjust the classifier prediction threshold based on the environment they were playing in. See the supporting information file for an image of this front-end user interface (S8 Appendix). Families were made aware of this feature in an onboarding session, however, it was not used in this study. Families did not experience dissatisfaction with the audio detection interface, so it may be that they did not feel a need for this feature. Future work will be conducted to quantify the extent to which specific environmental conditions impact accuracy and the effectiveness of mitigating strategies that can be designed into the technology or provided through training resources.

Another important consideration when thinking about target application was the relative consequence of false negatives (i.e., no detection of the instrument being played) versus false positives (i.e., detection of the instrument when it is not being played). While both may lead to a bad gameplay experience, young children tend to respond more negatively to false negatives. Thus, for our application, it was important to have a high recall and classify all instances of the instrument, even if that may lead to some additional false positives. In our case, our lowest recall in deployment was with the noise family with a median of 69.9% which was most often misclassified as a castanet. Upon further investigation, in most cases this was because the game audio had a castanet sound playing periodically in the background or because the family was moving around their instruments (creating a sharp onset and decay type of sound). However, a low recall for the noise class is not necessarily a bad thing as a false negative for noise means a false positive for an instrument class which is generally more acceptable than the counterpart. As noted above, families were also provided with a user interface to adjust the prediction threshold given that the optimal balance between false negatives and false positives may vary from child to child depending on motor abilities, self-efficacy and other individual characteristics. The families in this study did not make use of this feature and were satisfied with the audio detection interface.

Only one family noted any latency issues, and it was considered barely perceivable by the caregiver and unperceivable to the child. One reason for this is due to the game deployment itself. That is when Bootle Band prompts a user to play an instrument, it listens for an input for several windows. For example, Bootle Band may listen to a musical input for one second. In that second, with a ≈93 ms window and 50% overlap, inference is performed roughly 21 times. Therefore, even with an average performance of 73.3%, Bootle Band mitigates errors through gameplay programming.

In future work, we plan on expanding our dataset to include the real-world samples we recorded with our participants. Additionally, as Bootle Band is deployed commercially, it might be possible to collect a large variety of audio recordings that could be used for further quality improvement with the appropriate user permissions. It is common that machine learning models are frequently diagnosed and updated in deployment, and we see our classifier following a similar path which will hopefully lead to improved performance in real-world deployments. When more training data and real-world samples are available, we may also explore the performance of deep neural networks (DNNs) to further improve classification. DNN have been used successfully in widespread applications from speech-based emotion recognition [45–47] to music recognition [48] to detecting emotion in music [49]. Recent work with DNNs have shown immense potential for musical instrument classification as reviewed

by Blaszke and Kostek [50], particularly for predominant instrument recognition in polyphonic audio. To our knowledge, no previous work with DNNs have explored their application to the percussion instruments (maracas, tambourines, castanets) or sound environment (e.g. game music, children's play, home) of interest in this music application. While the classification accuracy obtained by the methods described herein appeared to provide children with a good user experience, DNNs might be a promising direction of exploration particularly if more instrument families are added to the system. As well reviewed by Blaszke and Kostek [50], the current state of the art for multiple instrument recognition yields F1 scores around 0.64 while their DNN approach provided substantial increases to 0.93 [50]. DNN approaches may also offer greater flexibility allowing for more complex models for instruments that are difficult to classify and simpler, more computationally efficient models for instruments that are easily identified [50]. The model architecture proposed by Blazke and Kostek also makes it possible to add new instruments [50]. Some challenges to using DNNs include the large datasets needed for training and computational costs [51].

In this work, we used a cut-off frequency of 8 kHz however we would like to explore the results of increasing our bandwidth to 20 kHz for our MFCC generation along with other signal processing techniques. Looking at the spectrograms for each instrument class, we expect that this might improve the misclassifications between shakers and tambourines. With the long-term goal of integrating our classifier into a low-cost, at-home music education application, we are also interested in the family experience (usability, engagement, perceived value) of mixed reality music applications like Bootle Band both on its own and when contrasted with typical touchscreen approaches. A parallel paper will describe the family experience with Bootle Band in this 2x2 crossover study design wherein children played Bootle Band with real-world instruments via the audio detection interface described herein or with virtual instruments via the touchscreen following which their experiences were captured through game logs, interviews and questionnaires. In the future, it would also be important to understand how the audio detection interface supports different game designs and play mechanics to understand how it can/should be deployed in applications. Working towards a music-making platform, we are interested in advanced pattern recognition to identify how well a child might keep a steady beat or reproduce a rhythmic pattern.

A final point of note: the goal of maximizing accessibility directed many of our design decisions, particularly with respect to: (i) the instruments included in the system, (ii) consideration of diverse play style, (iii) support of instrument variants and do-it-yourself instruments. The instruments we selected for our audio detection interface were selected through consultations with occupational and music therapists based on their prominence in early childhood musical play, and they also target different grasp and motor movements to play. In the early stages of development of our interface, we recognized that the sound characteristics of the musical instruments might vary quite significantly depending on the motor abilities of the child and how they were played (i.e., the play style). This was a key consideration when developing the database to train the instrument detection algorithms, as was the need to accommodate instrument variants (e.g., tambourines of different brands or fabrications that sound slightly different). The latter was to ensure that families could use instruments that they already might have on hand. To reduce potential economic constraints even further, we also included "do-it-yourself" instruments in the development of our algorithms. This design decision ensured that in the future, families would be able to participate in the musical experience using instruments fabricated from low-cost, household items. The art and crafts aspect to these homemade instruments provided an additional feature to the game which a few families reported to enjoy. It should be noted that future work would be needed to expand the audio detection interface to the wide range of non-pitched instruments that a child might encounter in the real-world as

this was not the focus of this first stage of research. We expect that the generalizability of our approach to new non-pitched percussion instruments would likely depend on the extent of overlap and similarity in sonic characteristics.

## Conclusion

In this work we developed a musical instrument classifier for non-pitched percussion instruments. By developing a non-pitched instrument classifier, we offer insight into how manipulation of different variables (i.e., window length, feature extraction, and model selection) contribute to the performance of a musical instrument classifier. We evaluated our classifier against in-lab variants and see accuracies over 84% using a $\approx$93 ms windowing time. Also, we deployed our classifier to 12 families within a music application called Bootle Band and see over 73% accuracy across all instrument families. Our classifier is the first step toward designing a low-cost at home music-making platform for children in early childhood. Ultimately, we expect that these results will help eliminate barriers that children and their families face such as: scheduling time for musical play and learning, the cost of musical play and learning, and the availability of a music program that is suitable to their children's abilities.

## Supporting information

**S1 Appendix. Do It Yourself (DIY) instruments.**
(PDF)

**S2 Appendix. List of feature extraction.** Italicized indicates selected features from NCA.
(PDF)

**S3 Appendix. Spectrogram of castanet (top), tambourine (middle) and shaker (bottom).**
Parameters: 44.1 kHz sampling rate, 50% overlap Hanning window, and 4096 samples DFT.
(PDF)

**S4 Appendix. Optimized LGBM parameters using Optuna.**
(PDF)

**S5 Appendix. Comparison of model performance on in-lab test set.** Using approximately 93 ms window and reporting macro-average result across all classes. Bold represents best performance.
(PDF)

**S6 Appendix. Confusion matrix analysis for the LGBM model using a 93ms window.**
(DOCX)

**S7 Appendix. SHAP results.** 'Class 0' = tambourines, 'Class 1' = shakers, 'Class 2' = castanets, 'Class 3' = noise.
(PDF)

**S8 Appendix. Bootle Band user interface for probability threshold of LGBM model.**
(PDF)

## Acknowledgments

The authors would like to thank the Institute of Biomedical Engineering at the University of Toronto, Holland Bloorview Kids Rehabilitation Hospital, and the Bloorview Research Institute for allowing us to conduct our research. The authors would like to thank Alexander Hodge for supporting integration within the Bootle Band app as well as Nikki Ponte and

Dorothy Choi who supported the real-world data collection. The authors would like to thank Dr. Azadeh Kushki for her insights and helping to shape this project.

## Author Contributions

**Conceptualization:** Brandon Rufino, Tilak Dutta, Elaine Biddiss.

**Data curation:** Brandon Rufino.

**Formal analysis:** Brandon Rufino, Ajmal Khan, Elaine Biddiss.

**Funding acquisition:** Brandon Rufino, Elaine Biddiss.

**Investigation:** Brandon Rufino, Elaine Biddiss.

**Methodology:** Brandon Rufino, Ajmal Khan, Tilak Dutta, Elaine Biddiss.

**Project administration:** Brandon Rufino.

**Resources:** Brandon Rufino, Tilak Dutta, Elaine Biddiss.

**Software:** Brandon Rufino, Ajmal Khan, Elaine Biddiss.

**Supervision:** Tilak Dutta, Elaine Biddiss.

**Validation:** Brandon Rufino.

**Visualization:** Brandon Rufino.

**Writing – original draft:** Brandon Rufino.

**Writing – review & editing:** Ajmal Khan, Tilak Dutta, Elaine Biddiss.

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
