## [Decision Letter · Decision Letter 0]

21 Dec 2023

PONE-D-23-33875Musical Instrument Classifier for Early Childhood Percussion InstrumentsPLOS ONE

Dear Dr. Biddiss,

Thank you for submitting your manuscript to PLOS ONE. After careful consideration, we feel that it has merit but does not fully meet PLOS ONE’s publication criteria as it currently stands. Therefore, we invite you to submit a revised version of the manuscript that addresses the points raised during the review process.

**ACADEMIC EDITOR:** Please revise and resubmit your manuscript. The requested citations in the review comments are not a requirement for publication. 

We look forward to receiving your revised manuscript.

Kind regards,

Kathiravan Srinivasan

Academic Editor

PLOS ONE

“I have read the journal's policy and the authors of this manuscript have the following competing interests: Holland Bloorview is supporting the creation of a company called Pearl Interactives to commercialize products like Bootle Band so that it can be made widely available to those who can benefit from it. Elaine Biddiss and Ajmal Khan are shareholders in Pearl Interactives and may financially benefit from this interest if Pearl Interactives is successful in marketing products related to this research including Bootle Band. The terms of this arrangement have been reviewed and approved by Holland Bloorview Kids Rehabilitation Hospital and the University of Toronto in accordance with its policy on objectivity in research. We will continue to actively monitor, mitigate and manage any conflicts of interest. Our goal is to remain transparent and committed to the best interests of study participants, patients and families.”

3. In this instance it seems there may be acceptable restrictions in place that prevent the public sharing of your minimal data. However, in line with our goal of ensuring long-term data availability to all interested researchers, PLOS’ Data Policy states that authors cannot be the sole named individuals responsible for ensuring data access (http://journals.plos.org/plosone/s/data-availability#loc-acceptable-data-sharing-methods).

5. We note that Figures S5 and S6 and S6 in your submission contain copyrighted images. All PLOS content is published under the Creative Commons Attribution License (CC BY 4.0), which means that the manuscript, images, and Supporting Information files will be freely available online, and any third party is permitted to access, download, copy, distribute, and use these materials in any way, even commercially, with proper attribution. For more information, see our copyright guidelines: http://journals.plos.org/plosone/s/licenses-and-copyright.

1. You may seek permission from the original copyright holder of Figure(s) [#] to publish the content specifically under the CC BY 4.0 license.

Reviewers' comments:

Reviewer's Responses to Questions

**Comments to the Author**

1. Is the manuscript technically sound, and do the data support the conclusions?

Reviewer #1: Yes

Reviewer #2: Yes

Reviewer #3: Partly

2. Has the statistical analysis been performed appropriately and rigorously? 

Reviewer #1: No

Reviewer #2: Yes

Reviewer #3: Yes

3. Have the authors made all data underlying the findings in their manuscript fully available?

Reviewer #1: Yes

Reviewer #2: No

Reviewer #3: No

4. Is the manuscript presented in an intelligible fashion and written in standard English?

Reviewer #1: Yes

Reviewer #2: Yes

Reviewer #3: No

5. Review Comments to the Author

Reviewer #1: 1. Please add paper contributions in the abstract.

2. Please add paper organization at the end of the introduction.

3. Please show the data distribution graph.

4. The paper didn’t consider the related work section.

5. Please add the following reference article for your discussion.

a. Music recommender system via deep learning, Journal of Information and Optimization Sciences 43 (5), 1081-1088.

6. A comparison of other machine learning models is missing.

7. State-of-the-art comparison is not considered in the analysis.

8. Confusion matrix analysis is missing.

9. Computation time analysis is not considered.

Reviewer #2: Decision: Major Revision

1. The paper mentions a dataset of diverse audio samples. How representative is this dataset of the wide range of non-pitched percussion instruments that children might encounter in real-world scenarios?

2. While the paper acknowledges challenges related to variability in home environments, the extent to which background noise and interruptions affect the classifier's performance could be explored in more detail. Are there specific environmental conditions that significantly impact accuracy?

3. The paper highlights the reduction in performance during real-world deployment. Could the paper delve deeper into the limitations of the model and potential causes for the decrease in accuracy?

4. The paper focuses on non-pitched percussion instruments. How generalizable is the proposed approach to other musical instruments, especially those with different sonic characteristics?

5. While the paper mentions the user interface feature allowing families to adjust the probability threshold, it could provide more insights into the effectiveness of this feature. Were families able to use it effectively, and did it enhance the overall user experience?

6. In the future work section, the paper mentions plans to expand the dataset. What specific strategies will be employed to ensure a more exhaustive dataset, and how might this impact the classifier's performance?

7. The paper suggests exploring deep neural networks in the future. Could the authors elaborate on the anticipated advantages and challenges of using deep learning for this particular classification task?

8. The paper mentions a parallel study to describe the family experience. Could the authors provide a brief overview of the methodology used to assess the family experience and its integration with the classifier's performance?

9. The paper discusses the consequence of false negatives versus false positives. How was the balance between minimizing false negatives and managing false positives determined, and what impact did this have on the overall user experience?

10. The manuscript, however, does not link well with recent literature on recognition appeared in relevant top-tier journals, e.g., the IEEE Intelligent Systems department on AAD-Net: Advanced end-to-end signal processing system for human emotion detection & recognition using attention-based deep echo state network". Also, new trends of AI for recognition “ARTriViT: Automatic Face Recognition System Using ViT-Based Siamese Neural Networks with a Triplet Loss” are missing it should be comprised.

11. Why is the proposed approach suitable to be used to solve the critical problem? We need more convinced response to indicate clearly the SOTA development.

12. Extension to literature will be appreciated: 10.32604/csse.2023.037373, 10.1109/BioSMART58455.2023.10162089. Please cite in section II behind the CNN and DL to enrich the literature.

Reviewer #3: The authors present a model for the identification of non-pitched percussion musical instruments, with a focus on instruments that are used in childhood education.

The presented investigation is meant to be used as a backbone and proof of concept for a target mixed reality application.

In the introduction, I am missing a final paragraph describing the structure of the rest of the paper, that is usually present and useful to the reader.

In my opinion Methods should be changed to Methodology or Materials and Methods.

One weak point of the paper is the references and citations. Most of the papers are very old, which does not look good for a paper in a rapidly evolving field, like machine learning for audio.

The extended description of the MFCCs and entropy features is not in my opinion needed in a research paper, since the aim of the contribution is not there, but in the novelty aspect.

My main concern regarding the methodology is the window length. First of all, [23] is used as a reference for the selection of window, but it is related to a much different task (music transcription). In [44], even though a small window is used for MFCC extraction (1024samples), classification is performed for 4second frames.

Another thing that I suggest should be improved is the structure of the paper. First of all, there is a plan to use the classifier in a mixed reality application. This is mentioned several times, in different places. I believe that there should be a section or sub-section clarifying what kind of game this is, etc. It is difficult for the reader to follow the concept, when different parts are mixed up.

I feel the same about dataset creation. There is a section for the dataset creation of the classifier, and there is another section concerning an experiment with the classifier used within a game.

I believe it is an interesting work but needs rewriting with emphasis on:

1) Defining the structure of the paper in the introduction

2) Making more clear what kind of application is the task expected to be used in

3) What kind of experiments are held within the paper.

4) An update of the references with more recent works

5) A better justification of the window length, which seems quite small. Can humen identify the instruments in such a length? What is the length of the recordings? For example, how many ms is approximately one hit of the maraca or tambourine. Is one hit broken into smaller samples in the proposed methodology? Is it possible that samples that belong to the same hit are present in both training and testing?

6) Has there been some kind of normalization of audio samples? Classifiers can adapt to different energy levels if the dataset is not normalized. I believe that the dataset is one of the strong points of the paper and it should be presented more clearly. Is it available? This could also strengthen both the contribution of the paper and also the transparency.

I understand that many of these questions maybe answered, but I believe that the reader should be able to find the information they want without having to carefully read the whole paper, and this is why I propose a better structure.

7) I don't understand the relevance of the demographics

6. PLOS authors have the option to publish the peer review history of their article (what does this mean?). If published, this will include your full peer review and any attached files.

Reviewer #1: **Yes: **

Reviewer #2: No

Reviewer #3: No

---

## [Author Response · Author response to Decision Letter 0]

4 Feb 2024

Editorial comments: 

As requested, the PLOS ONE style templates were consulted and corresponding changes were made to adhere to PLOS ONE’s style requirements.

2. Thank you for stating the following in the Competing Interests section. Please confirm that this does not alter your adherence to all PLOS ONE policies on sharing data and materials, by including the following statement: "This does not alter our adherence to PLOS ONE policies on sharing data and materials.” Please include your updated Competing Interests statement in your cover letter; we will change the online submission form on your behalf.

As requested, we have added the updated Competing Interests statement in our cover letter.

3. In this instance it seems there may be acceptable restrictions in place that prevent the public sharing of your minimal data. However, in line with our goal of ensuring long-term data availability to all interested researchers, PLOS’ Data Policy states that authors cannot be the sole named individuals responsible for ensuring data access (http://journals.plos.org/plosone/s/data-availability#loc-acceptable-data-sharing-methods).

Before we proceed with your manuscript, please also provide non-author contact information (phone/email/hyperlink) for a data access committee, ethics committee, or other institutional body to which data requests may be sent. Please also provide details on how you will ensure persistent or long-term data storage and availability.

As requested, our revised data availability statement is as follows:

“The data that support the findings of this study may be made available on request from the corresponding author, E.B, in compliance with institutional and ethical standards of operation. Data cannot be shared publicly because research participants did not provide consent for public sharing of their data. To ensure the long-term stability and accessibility of our research data, we will designate a non-author institutional contact, the research ethics committee chair. This approach ensures that the data remains accessible over time, providing a reliable point of contact for interested researchers. Such an arrangement is particularly beneficial in cases where an author may change their email address, shift to a different institution, or become unavailable to respond to data access requests. Please see the contact information for the non-author institutional contact below:

Deryk Beal

Research Ethics Board Chair 

Holland Bloorview Kids Rehabilitation Hospital 

150 Kilgour Road, Toronto, ON M4G 1R8 

Tel: (416) 425-6220, ext.3582 

E-mail: dbeal@hollandbloorview.ca”

Our full ethics statement is mentioned in the ‘Methods’ section of our manuscript file as follows: “Ethical approval for this study was obtained from the Holland Bloorview Research Ethics Board and the University of Toronto Health Sciences Research Ethics Board (eREB project ID #0257). Participants and/or their guardians provided informed and written consent using approved e-consent procedures facilitated by REDCap. Children who were unable to consent provided assent.”

5. We note that Figures S5 and S6 in your submission contain copyrighted images. All PLOS content is published under the Creative Commons Attribution License (CC BY 4.0), which means that the manuscript, images, and Supporting Information files will be freely available online, and any third party is permitted to access, download, copy, distribute, and use these materials in any way, even commercially, with proper attribution. We require you to either (1) present written permission from the copyright holder to publish these figures specifically under the CC BY 4.0 license, or (2) remove the figures from your submission:

As requested, we have uploaded the completed Content Permission Form as an "Other" file with our submission. In the figure caption of the copyrighted figures, we’ve include the following text: “Reprinted from Pearl Interactives under a CC BY license, with permission from Sharon Wong, CEO of Pearl Interactives, original copyright 2024.”

Reviewer comments:

Reviewer #1: 

1. Please add paper contributions in the abstract.

As requested, we have added the following paper contributions to the abstract as follows:

“To our knowledge, the dataset compiled of 369,000 samples of non-pitched instruments is first of its kind. This work also suggests that a low feature space is sufficient for the recognition of non-pitched instruments. Lastly, real-world deployment and testing of the algorithms created with participants of diverse physical and cognitive abilities was also an important contribution towards more inclusive design practices. This paper lays the technological groundwork for a mixed reality music application that can detect children’s use of non-pitched, percussion instruments to support early childhood music education and play.”

2. Please add paper organization at the end of the introduction.

As requested, we have added a description of the paper organization to the end of the introduction. Thank you, we think this improves the paper’s readability:

“In this work, we aimed to develop an audio detection interface for non-pitched percussion instruments, specifically maracas, tambourines, and castanets, for use in early childhood music applications. To this end, this manuscript first describes the creation of a large database of non-pitched instrument audio samples. Second, we describe feature extraction and the development of machine learning models used in this classification task. Next, we present the performance of our classifier with (i) a test set recorded in-lab and, (ii) real-world data recorded “in the wild” in family homes. To collect the latter, the audio detection interface was deployed in a music application called Bootle Band with 12 families with children of diverse abilities. Bootle Band is an early childhood music application developed at the Possibility Engineering And Research Lab (PEARL) at Holland Bloorview Kids Rehabilitation Hospital and available on the Apple App Store. Lastly, we discuss key findings, particularly with respect to the algorithms intended application as an audio detection interface to support interactive early childhood music applications.” 

3. Please show the data distribution graph.

We have included the data distribution with respect to the instruments and data splits as Table 1. Please let us know if you would like additional information and what would be helpful and we are happy to provide. Thank you.

4. The paper didn’t consider the related work section.

The related work section has been expanded in the introduction section as follows:

Thank you for this suggestion. We have expanded the related work in the introduction section as follows:

“Musical instrument and audio classifiers have been built with a wide range of machine learning models including k-nearest neighbors (KNN) [35], Multi-layered Perceptron (MLP), and boosting algorithms [36]. Harish et al. reported accuracies of 79% for their SVM model which outperformed the other state-of-the-art models with in classifying six pitched instruments, including voice, using spectral features [36]. Mittal et al. demonstrated best performance with a Naive Bayes Classifier with an accuracy of 97% for distinguishing between 4 drum instruments using a dataset composed of both live recordings as well as a drum simulator [37]. While the musical instrument classification task for pitched instruments and drums are widely studied [10], we were not able to find previous work classifying diverse non-pitched instruments like maracas, tambourines, and castanets. Classification of non-pitched instruments may pose additional challenges due to greater overlaps in frequency bands and variation in sound quality and play style than pitched instruments [14]. “

We have also included greater discussion of deep learning techniques in the discussion as follows:

 “As well reviewed by Blaszke and Kostek [50], the current state of the art for multiple instrument recognition yields F1 scores around 0.64 while their Deep Neural Nets (DNN) approach provided substantial increases to 0.93 [50]. DNN approaches may also offer greater flexibility allowing for more complex models for instruments that are difficult to classify and simpler, more computationally efficient models for instruments that are easily identified [50]. The model architecture proposed by Blazke and Kostek also makes it possible to add new instruments [50]. Some challenges to using DNNs include the large datasets needed for training and computational costs [51].”

51) Zaman K, Sah M, Direkoglu C, Unoki M. A Survey of Audio Classification Using Deep Learning. IEEE Access. 2023 Sep 22.

5. Please add the following reference article for your discussion.

a. Music recommender system via deep learning, Journal of Information and Optimization Sciences 43 (5), 1081-1088.

The requested article has been added to the discussion section as requested.

6. A comparison of other machine learning models is missing.

In agreement and as requested, the comparison to other machine learning models is provided in the supporting file (S5) as follows:

S5 Appendix. Comparison of model performance on in-lab test set using approximately 93 ms window and reporting macro-average result across all classes. Bold represents best performance.

Please let us know if you prefer for the comparison table to appear in the main manuscript as opposed to the supporting information.

7. State-of-the-art comparison is not considered in the analysis.

The state-of-the-art comparison is now provided in the Results (In-lab Performance) analysis as follows: 

“The LGBM classifier performed best on our validation set across all window lengths and for all performance metrics (Precision: 0.845; Recall: 0.835, F1: 0.839, Accuracy: 0.844, 93ms window). The AdaBoost model had similar performance (Accuracy: 0.837, 93ms window) with the logistic regression model (Accuracy: 0.676, 93ms window) and the SVM (Accuracy: 0.756, 93ms window) being the lowest performing.”

8. Confusion matrix analysis is missing.

As requested, we have added a confusion matrix to the supporting files (S6)

Please let us know if you prefer for the comparison table to appear in the main manuscript as opposed to the supporting information.

This was an important addition to support our discussion of the relative impact of false negatives and false positives provided in the discussion section as follows:

“Another important consideration when thinking about target application was the relative consequence of false negatives (i.e., no detection of the instrument being played) versus false positives (i.e., detection of the instrument when it is not being played). While both may lead to a bad gameplay experience, young children tend to respond more negatively to false negatives. Thus, for our application, it was important to have a high recall and classify all instances of the instrument, even if that may lead to some additional false positives. In our case, our lowest recall in deployment was with the noise family with a median of 69.9% which was most often misclassified as a castanet. Upon further investigation, in most cases this was because the game audio had a castanet sound playing periodically in the background or because the family was moving around their instruments (creating a sharp onset and decay type of sound). However, a low recall for the noise class is not necessarily a bad thing as a false negative for noise means a false positive for an instrument class which is generally more acceptable than the counterpart.”

9. Computation time analysis is not considered. 

The model results using different window lengths is now provided in the Results in the “In-Lab performance” section as follows:

“Table 3 presents the performance of the LGBM model on the held-out “in-lab” test set. Using a ≈23 ms (1024 sample) window length we see an overall accuracy of about 72.6%. This accuracy increases when we expand our window to ≈46 ms (2048 samples) and ≈93 ms (4096 samples). Respectively, we see an increase in accuracy of 1.4% (to a total of 74.0%) and an increase of 11.8% (to a total of 84.4%) from our smallest window size of ≈23 ms. Another substantial increase in performance can be observed in the precision of our classifier. For example, the lowest precision score was in the Tambourine class with a ≈23 ms window at 67.5%. This precision improved to 86.7% when the window size was increased to 93 ms.” 

Table 3. Non-pitched percussion instrument classifier to our “in-lab” test set against different window lengths.

We also provide results pertaining to the families’ perception of the computation time in the Results under the section heading, Real-World Deployment Performance, as follows:

“To coincide with the quantitative analysis, when families were asked “Did you experience any technical difficulties with the detection of musical instruments?”, eleven out of twelve families reported no latency or detection issues with Bootle Band. One out of the twelve caregivers noticed a very minor delay when an instrument was played vs. the reaction time of the game. However, the caregiver noted that their child never seemed to notice and it did not impede the play experience.”

Finally, in the discussion, we address computation time as follows:

“Good classifier performance with low latency and computational demands typical of common mobile devices could be achieved using a small feature set. We found that only a handful of features (MFCCs and signal entropy) were needed to classify percussion instruments in a short window of time (≈93 ms). By minimizing our features, we were able to reduce bias in our model and minimize computation time. This is crucial for our target application which required latency of less than 100 ms to ensure a good user experience [24]. While larger window times were associated with an increase in classifier performance, it is unlikely that this would have translated to a better user experience when embedded into a gaming application.”

We also included in the discussion section, some notes regarding the deployment of the algorithm in the Bootle Band app and how that would have impacted the user’s perception of the computation time: 

“Only one family noted any latency issues, and it was considered barely perceivable by the caregiver and unperceivable to the child. One reason for this is due to the game deployment itself. That is when Bootle Band prompts a user to play an instrument, it listens for an input for several windows. For example, Bootle Band may listen to a musical input for one second. In that second, with a ≈93 ms window and 50% overlap, inference is performed roughly 21 times. Therefore, even with an average performance of 73.3%, Bootle Band mitigates errors through gameplay programming.”

Reviewer #2: Decision: Major Revision

1. The paper mentions a dataset of diverse audio samples. How representative is this dataset of the wide range of non-pitched percussion instruments that children might encounter in real-world scenarios?

Thank you for this comment. It is an important one. The dataset was created not with the intent of representing every non-pitched percussion that a child might encounter in the real-world. Rather, it was designed and instruments were selected with a focus on accessibility. To address this, we have added the following to the discussion section: 

“The goal of maximizing accessibility directed many of our design decisions, particularly with respect to: (i) the instruments included in the system, (ii) consideration of diverse play style, (iii) support of instrument variants and do-it-yourself instruments. The instruments we selected for our audio detection interface were selected through consultations with occupational and music therapists based on their prominence in early childhood musical play, and they also target different grasp and motor movements to play. In the early stages of development of our interface, we recognized that the sound characteristics of the musical instruments might vary quite significantly depending on the motor abilities of the child and how they were played (i.e., the play style). This was a key consideration when developing the database to train the instrument detection algorithms, as was the need to accommodate instrument variants (e.g., tambourines of different brands or fabrications that sound slightly different). The latter was to ensure that families could use instruments that they already might have on hand. To reduce potential economic constraints even further, we also included “do-it-yourself” instruments in the development of our algorithms. This design decision ensured that in the future, families would be able to participate in the musical experience using instruments fabricated from low-cost, household items. The art and crafts aspect to these homemade instruments provided an additional feature to the game which a few families reported to enjoy. It should be noted that future work would be needed to expand the audio detection interface to the wide range of non-pitched instruments that a child might encounter in the real-world as this was not the focus of this first stage of research.” 

2. While the paper acknowledges challenges related to variability in home environments, the extent to which background noise and interruptions affect the classifier's performance could be explored in more detail. Are there specific environmental conditions that significantly impact accuracy?

In response to the reviewer, the following is included in the discussion section:

“In the home setting, misclassifications could be attributed to many different events: children yelling, the microphone being covered when a child holds the iPad, and children playing two different instruments at the same time with their sibling/parent. Upon exploratory investigation we noticed that when the child was playing Bootle Band in a quiet environment without many interruptions like the ones listed above, the classifier performed with near equivalence to in-lab testing. Understanding the source of errors enables opportunities to provide families with better instructions for an optimal user experience as well as to inform technological developments that could help to mitigate misclassification (e.g., programmatic volume controls associated with the game music, warnings if the microphone is covered). As an example of the latter, one feature that we implemented to support deployment is that we allowed users, through a friendly front-end user interface, to toggle the probability threshold of predicting an instrument. For example, if a family was playing in a loud environment, then to reduce the number of false positives we allowed them to increase the probability threshold of our classifier to a max of 0.9. If the user were playing in a quiet environment, they could lower the probability threshold to 0.4. Doing this allowed the family to adjust the classifier prediction threshold based on the environment they were playing in. See the supporting information file for an image of this front-end user interface (S8). Families were made aware of this feature in an onboarding session, however, it was not used in this study. Families did not experience dissatisfaction with the audio detection interface, so it may be that they did not feel a need for this feature. Future work will be conducted to quantify the extent to which specific environmental conditions impact accuracy and the effectiveness of mitigating strategies that can be designed into the technology or provided through training resources as needed.”

3. The paper highlights the reduction in performance during real-world deployment. Could the paper delve deeper into the limitations of the model and potential causes for the decrease in accuracy?

The following is provided in the Discussion section pertaining to the reduction in performance in the real-world deployment.

“Our classifier achieved 84.4% accuracy across all three instrument families when evaluated with our in-lab test set with ≈93 ms windows. In deployment, it achieved 73.3% accuracy across all three instrument families. In the home setting, misclassifications could be attributed to many different events: children yelling, the microphone being covered when a child holds the iPad, and children playing two different instruments at the same time with their sibling/parent. Upon exploratory investigation we noticed that when the child was playing Bootle Band in a quiet environment without many interruptions like the ones listed above, the classifier performed with near equivalence to in-lab testing.” 

4. The paper focuses on non-pitched percussion instruments. How generalizable is the proposed approach to other musical instruments, especially those with different sonic characteristics?

In response, we have added the following to the discussion of limitations. Thank you.

“It should be noted that future work would be needed to expand the audio detection interface to the wide range of non-pitched instruments that a child might encounter in the real-world as this was not the focus of this first stage of research. We expect that the generalizability of our approach to new non-pitched percussion instruments would likely depend on the extent of overlap and similarity in sonic characteristics.” 

5. While the paper mentions the user interface feature allowing families to adjust the probability threshold, it could provide more insights into the effectiveness of this feature. Were families able to use it effectively, and did it enhance the overall user experience?

Thank you for your interest in this feature. As none of the families experienced dissatisfaction with the detection interface, this feature was not used. This clarification has been added to the manuscript as follows:

“Families were made aware of this feature in an onboarding session, however, it was not used in this study. Families did not experience dissatisfaction with the audio detection interface, so it may be that they did not feel a need for this feature.”

6. In the future work section, the paper mentions plans to expand the dataset. What specific strategies will be employed to ensure a more exhaustive dataset, and how might this impact the classifier's performance?

Thank you. In response, our plan for expanding the dataset is described in the discussion as follows:

“In future work, we plan on expanding our dataset to include the real-world samples we recorded with our participants. Additionally, as Bootle Band is deployed commercially, it might be possible to collect a large variety of audio recordings that could be used for further quality improvement with the appropriate user permissions. It is common that machine learning models are frequently diagnosed and updated in deployment, and we see our classifier following a similar path which will hopefully lead to improved performance in real-world deployments.”

7. The paper suggests exploring deep neural networks in the future. Could the authors elaborate on the anticipated advantages and challenges of using deep learning for this particular classification task?

We have added further discussion as requested by the reviewer as follows:

“As well reviewed by Blaszke and Kostek [50], the current state of the art for multiple instrument recognition yields F1 scores around 0.64 while their DNN approach provided substantial increases to 0.93 [50]. DNN approaches may also offer greater flexibility allowing for more complex models for instruments that are difficult to classify and simpler, more computationally efficient models for instruments that are easily identified [50]. The model architecture proposed by Blazke and Kostek also makes it possible to add new instruments [50]. Some challenges to using DNNs include the large datasets needed for training and computational costs [51].”

51) Zaman K, Sah M, Direkoglu C, Unoki M. A Survey of Audio Classification Using Deep Learning. IEEE Access. 2023 Sep 22.

8. The paper mentions a parallel study to describe the family experience. Could the authors provide a brief overview of the methodology used to assess the family experience and its integration with the classifier's performance?

Thank you for your interest. This study has been described in better detail under the Discussion as follows:

“A parallel paper will describe the family experience with Bootle Band in this 2x2 crossover study design wherein children played Bootle Band with real-world instruments via the audio detection interface described herein or with virtual instruments via the touchscreen following which their experiences were captured through game logs, interviews and questionnaires. “

9. The paper discusses the consequence of false negatives versus false positives. How was the balance between minimizing false negatives and managing false positives determined, and what impact did this have on the overall user experience?

We have included the following discussion with respect to false negatives and false positives and the overall user experience in the discussion

“Another important consideration when thinking about target application was the relative consequence of false negatives (i.e., no detection of the instrument being played) versus false positives (i.e., detection of the instrument when it is not being played). While both may lead to a bad gameplay experience, young children tend to respond more negatively to false negatives. Thus, for our application, it was important to have a high recall and classify all instances of the instrument, even if that may lead to some additional false positives. In our case, our lowest recall in deployment was with the noise family with a median of 69.9% which was most often misclassified as a castanet. Upon further investigation, in most cases this was because the game audio had a castanet sound playing periodically in the background or because the family was moving around their instruments (creating a sharp onset and decay type of sound). However, a low recall for the noise class is not necessarily a bad thing as a false negative for noise means a false positive for an instrument class which is generally more acceptable than the counterpart. As noted above, families were also provided with a user interface to adjust the prediction threshold given that the optimal balance between false negatives and false positives may vary from child to child depending on motor abilities, self-efficacy and other individual characteristics. The families in this study did not make use of this feature and were satisfied with the audio detection interface.”

10. The manuscript, however, does not link well with recent literature on recognition appeared in relevant top-tier journals, e.g., the IEEE Intelligent Systems department on AAD-Net: Advanced end-to-end signal processing system for human emotion detection & recognition using attention-based deep echo state network". Also, new trends of AI for recognition “ARTriViT: Automatic Face Recognition System Using ViT-Based Siamese Neural Networks with a Triplet Loss” are missing it should be comprised.

We have added some of the suggested citations to the discussion as suggested.

11. Why is the proposed approach suitable to be used to solve the critical problem? We need more convinced response to indicate clearly the SOTA development.

 Thank you for this comment. We have expanded the related work in the introduction section as follows:

“Musical instrument and audio classifiers have been built with a wide range of machine learning models including k-nearest neighbors (KNN) [35], Multi-layered Perceptron (MLP), and boosting algorithms [36]. Harish et al. reported accuracies of 79% for their SVM model which outperformed the other state-of-the-art models with in classifying six pitched instruments, including voice, using spectral features [36]. Mittal et al. demonstrated best performance with a Naive Bayes Classifier with an accuracy of 97% for distinguishing between 4 drum instruments using a dataset composed of both live recordings as well as a drum simulator [37]. While the musical instrument classification task for pitched instruments and drums are widely studied [10], we were not able to find previous work classifying diverse non-pitched instruments like maracas, tambourines, and castanets. Classification of non-pitched instruments may pose additional challenges due to greater overlaps in frequency bands and variation in sound quality and play style than pitched instruments [14]. “

We have also included greater discussion of deep learning techniques in the discussion as follows:

 “As well reviewed by Blaszke and Kostek [50], the current state of the art for multiple instrument recognition yields F1 scores around 0.64 while their Deep Neural Nets (DNN) approach provided substantial increases to 0.93 [50]. DNN approaches may also offer greater flexibility allowing for more complex models for instruments that are difficult to classify and simpler, more computationally efficient models for instruments that are easily identified [50]. The model architecture proposed by Blazke and Kostek also makes it possible to add new instruments [50]. Some challenges to using DNNs include the large datasets needed for training and computational costs [51].”

51) Zaman K, Sah M, Direkoglu C, Unoki M. A Survey of Audio Classification Using Deep Learning. IEEE Access. 2023 Sep 22.

Working off the prior work we shared above. We feel our contributions bring several unique differences. These are now stated in our abstract:

“To our knowledge, the dataset compiled of 369,000 samples of non-pitched instruments is first of its kind. This work also suggests that a low feature space is sufficient for the recognition of non-pitched instruments. Lastly, real-world deployment and testing of the algorithms created with participants of diverse physical and cognitive abilities was also an important contribution towards more inclusive design practices. This paper lays the technological groundwork for a mixed reality music application that can detect children’s use of non-pitched, percussion instruments to support early childhood music education and play.”

12. Extension to literature will be appreciated: 10.32604/csse.2023.037373, 10.1109/BioSMART58455.2023.10162089. Please cite in section II behind the CNN and DL to enrich the literature.

As requested, we have included the suggested citation.

Reviewer #3: 

In the introduction, I am missing a final paragraph describing the structure of the rest of the paper, that is usually present and useful to the reader.

We agree and have added the requested paragraph as follows:

“In this work, we aimed to develop an audio detection interface for non-pitched percussion instruments, specifically maracas, tambourines, and castanets, for use in early childhood music applications. To this end, this manuscript first describes the creation of a large database of non-pitched instrument audio samples. Second, we describe feature extraction and the development of machine learning models used in this classification task. Next, we present the performance of our classifier with (i) a test set recorded in-lab and, (ii) real-world data recorded “in the wild” in family homes. Lastly, we discuss key findings, particularly with respect to the algorithms intended application as an audio detection interface to support interactive early childhood music applications.” 

In my opinion Methods should be changed to Methodology or Materials and Methods.

We agree and have changed the heading to Methodology

One weak point of the paper is the references and citations. Most of the papers are very old, which does not look good for a paper in a rapidly evolving field, like machine learning for audio.

In agreement, we have redone the literature search and added more recent citations. This has been added to the Introduction section as follows:

“Musical instrument and audio classifiers have been built with a wide range of machine learning models including k-nearest neighbors (KNN) [35], Multi-layered Perceptron (MLP), and boosting algorithms [36]. Harish et al. reported accuracies of 79% for their SVM model which outperformed the other state-of-the-art models with in classifying six pitched instruments, including voice, using spectral features [36].Mittal et al. demonstrated best performance with a Naive Bayes Classifier with an accuracy of 97% for distinguishing between 4 drum instruments using a dataset composed of both live recordings as well as a drum simulator [37]. While the musical instrument classification task for pitched instruments and drums are widely studied [10], we were not able to find previous work classifying diverse non-pitched instruments like maracas, tambourines, and castanets. Classification of non-pitched instruments may pose additional challenges due to greater overlaps in frequency bands and variation in sound quality and play style than pitched instruments [14].”

We have also added some more recent literature to the discussion section as follows:

“DNN have been used successfully in widespread applications from speech-based emotion recognition [45-47] to music recognition [48] to detecting emotion in music [49]. Recent work with DNNs have shown immense potential for musical instrument classification as reviewed by Blaszke and Kostek [50]”

Citations added are as follows:

37) Chhabra A, Singh AV, Srivastava R, Mittal V. Drum Instrument Classification Using Machine Learning. In2020 2nd International Conference on Advances in Computing, Communication Control and Networking (ICACCCN) 2020 Dec 18 (pp. 217-221). IEEE.

45) Swain M, Maji B, Khan M, El Saddik A, Gueaieb W. Multilevel feature representation for hybrid transformers-based emotion recognition. In2023 5th International Conference on Bio-engineering for Smart Technologies (BioSMART) 2023 Jun 7 (pp. 1-5). IEEE.

46) Mustaqeem K, El Saddik A, Alotaibi FS, Pham NT. AAD-Net: Advanced end-to-end signal processing system for human emotion detection & recognition using attention-based deep echo state network. Knowledge-Based Systems. 2023 Jun 21;270:110525.

47) Ishaq M, Khan M, Kwon S. TC-Net: A Modest & Lightweight Emotion Recognition System Using Temporal Convolution Network. Computer Systems Science & Engineering. 2023 Sep 1;46(3).

48) Poulose A, Reddy CS, Dash S, Sahu BJ. Music recommender system via deep learning. Journal of Information and Optimization Sciences. 2022 Jul 4;43(5):1081-8.

49) Ramírez J, Flores MJ. Machine learning for music genre: multifaceted review and experimentation with audioset. Journal of Intelligent Information Systems. 2020 Dec;55(3):469-99.

The extended description of the MFCCs and entropy features is not in my opinion needed in a research paper, since the aim of the contribution is not there, but in the novelty aspect.

Thank you. We have reduced the description of the MFCCs and entropy features and have provided appropriate references for the interested reader to refer to if they require more computational details.

My main concern regarding the methodology is the window length. First of all, [23] is used as a reference for the selection of window, but it is related to a much different task (music transcription). In [44], even though a small window is used for MFCC extraction (1024samples), classification is performed for 4second frames. A better justification of the window length, which seems quite small. Can humen identify the instruments in such a length? What is the length of the recordings? For example, how many ms is approximately one hit of the maraca or tambourine. Is one hit broken into smaller samples in the proposed methodology? Is it possible that samples that belong to the same hit are present in both training and testing?

Thank you for this excellent point. The gaming literature suggests that “latency of less than 100 ms to ensure a good user experience [24]” which motivated our decision to use a window length no greater than 100ms. This is noted in the Methods section (Features and Window length).

As for your second point, it is an important one and we completely agree as test results could have been biased if the same instrument samples were within our training and testing set. We have added a sentence to the methods section (Dataset splits) to clarify that this was not the case.

“We labelled and grouped by instruments before data splitting to ensure that there was no overlap in samples between the training and testing sets. The test set fully comprised of either (1) instruments that were not included in training and validation, or (2) instruments that were included in training/validation but recorded in a totally new environment. This prevented the classifier from being biased to certain brands and materials of instruments.” 

Another thing that I suggest should be improved is the structure of the paper. First of all, there is a plan to use the classifier in a mixed reality application. This is mentioned several times, in different places. I believe that there should be a section or sub-section clarifying what kind of game this is, etc. It is difficult for the reader to follow the concept, when different parts are mixed up. I feel the same about dataset creation. There is a section for the dataset creation of the classifier, and there is another section concerning an experiment with the classifier used within a game.

Thank you. Upon re-read, we realize that the structure of the paper was not clear. We have reorganized the paper significantly and we hope this makes it an easier read. 

Has there been some kind of normalization of audio samples? Classifiers can adapt to different energy levels if the dataset is not normalized. I believe that the dataset is one of the strong points of the paper and it should be presented more clearly. Is it available? This could also strengthen both the contribution of the paper and also the transparency.

Normalization of the audio sample was carried out. The dataset is not yet available. We have not yet been able to secure the funding necessary (e.g. to ensure that it’s properly documented and for the home dataset, that there are no identifying information in the recordings).

I don't understand the relevance of the demographics

It is our institution’s policy to collect and report demographics fully (including socioeconomic status, ethnicity, gender, etc.) to promote equity, diversity, and inclusion in research. 

As requested, we have uploaded our figure files to the PACE to ensure that figures meet PLOS requirements. The changes made by PACE were the resolution being changed to 300 PPI, and the TIFF file being converted to a valid TIF file. We have uploaded the updated figure files to our revised submission.

---

## [Decision Letter · Decision Letter 1]

19 Feb 2024

Musical Instrument Classifier for Early Childhood Percussion Instruments

PONE-D-23-33875R1

Dear Dr. Biddiss,

We’re pleased to inform you that your manuscript has been judged scientifically suitable for publication and will be formally accepted for publication once it meets all outstanding technical requirements.

Kind regards,

John Blake, PhD

Academic Editor

PLOS ONE

Additional Editor Comments (optional):

Reviewers' comments:

Reviewer's Responses to Questions

**Comments to the Author**

1. If the authors have adequately addressed your comments raised in a previous round of review and you feel that this manuscript is now acceptable for publication, you may indicate that here to bypass the “Comments to the Author” section, enter your conflict of interest statement in the “Confidential to Editor” section, and submit your "Accept" recommendation.

Reviewer #1: All comments have been addressed

Reviewer #2: All comments have been addressed

2. Is the manuscript technically sound, and do the data support the conclusions?

Reviewer #1: Yes

Reviewer #2: Yes

3. Has the statistical analysis been performed appropriately and rigorously? 

Reviewer #1: Yes

Reviewer #2: Yes

4. Have the authors made all data underlying the findings in their manuscript fully available?

Reviewer #1: Yes

Reviewer #2: No

5. Is the manuscript presented in an intelligible fashion and written in standard English?

Reviewer #1: No

Reviewer #2: Yes

6. Review Comments to the Author

Reviewer #1: Dear Authors,

Thank you for addressing all my comments and I don't have any further concerns on your paper.

Regards

Reviewer #2: The authors successfully addressed my comments and suggestions. Good Luck!

The authors successfully addressed my comments and suggestions. Good Luck!

The authors successfully addressed my comments and suggestions. Good Luck!

7. PLOS authors have the option to publish the peer review history of their article (what does this mean?). If published, this will include your full peer review and any attached files.

Reviewer #1: **Yes: **Alwin Poulose

Reviewer #2: No

---

## [Editor Report · Acceptance letter]

22 Mar 2024

PONE-D-23-33875R1 

PLOS ONE

Dear Dr. Biddiss, 

I'm pleased to inform you that your manuscript has been deemed suitable for publication in PLOS ONE. Congratulations! Your manuscript is now being handed over to our production team.

Kind regards, 

on behalf of

Dr. John Blake 

Academic Editor

PLOS ONE